# Eocene intra-plate shortening responsible for the rise of a faunal pathway in the northeastern Caribbean realm

**Mélody Philippon**[1], **Jean-Jacques Cornée**[2], **Philippe Münch**[2], **Douwe J. J. van Hinsbergen**[3], **Marcelle BouDagher-Fadel**[4], **Lydie Gailler**[5], **Lydian M. Boschman**[6], **Frédéric Quillevere**[7], **Leny Montheil**[2], **Aurelien Gay**[2], **Jean Frédéric Lebrun**[1,2], **Serge Lallemand**[2], **Laurent Marivaux**[8], **Pierre-Olivier Antoine**[8], with the GARANTI Team[¶]

1 Geosciences Montpellier, Université de Montpellier, CNRS, Université des Antilles, Pointe-à-Pitre, France, 2 Geosciences Montpellier, Université de Montpellier, CNRS, Université des Antilles, Montpellier, France, 3 Department of Earth Sciences, Utrecht University, Utrecht, the Netherlands, 4 Office of the Vice-Provost (Research), University College London, London, United Kingdom, 5 Institut de Recherche pour le Developpement, Université Jean Monnet, Clermont-Ferrand, France, 6 Department of Environmental Systems Science, ETH Zürich, Zürich, Switzerland, 7 Univ. Lyon, Université Claude Bernard Lyon 1, ENS de Lyon, UMR 5276 LGL-TPE, Villeurbanne, France, 8 Institut des Sciences de l evolution de Montpellier (ISE-M, UMR 5554, CNRS/UM/IRD/EPHE), Laboratoire de Paléontologie, Montpellier, France

¶ GARANTI Team membership list can be found in the Acknowledgments section.
* melody.philippon@univ-antilles.fr

**Data Availability Statement:** All relevant data are within the manuscript and its Supporting Information files.

## Abstract

Intriguing latest Eocene land-faunal dispersals between South America and the Greater Antilles (northern Caribbean) has inspired the hypothesis of the GAARlandia (Greater Antilles Aves Ridge) land bridge. This landbridge, however, should have crossed the Caribbean oceanic plate, and the geological evolution of its rise and demise, or its geodynamic forcing, remain unknown. Here we present the results of a land-sea survey from the northeast Caribbean plate, combined with chronostratigraphic data, revealing a regional episode of mid to late Eocene, trench-normal, E-W shortening and crustal thickening by ∼25%. This shortening led to a regional late Eocene–early Oligocene hiatus in the sedimentary record revealing the location of an emerged land (the Greater Antilles-Northern Lesser Antilles, or GrANoLA, landmass), consistent with the GAARlandia hypothesis. Subsequent submergence is explained by combined trench-parallel extension and thermal relaxation following a shift of arc magmatism, expressed by a regional early Miocene transgression. We tentatively link the NE Caribbean intra-plate shortening to a well-known absolute and relative North American and Caribbean plate motion change, which may provide focus for the search of the remaining connection between 'GrANoLA' land and South America, through the Aves Ridge or Lesser Antilles island arc. Our study highlights the how regional geodynamic evolution may have driven paleogeographic change that is still reflected in current biology.

## Introduction

Darwin [1] already recognized that the West Indies (Greater Antilles, Bahamas, and Lesser Antilles) appear to have been colonized by South American terrestrial mammal faunas. By

**Funding:** Financial support has been provided by INSU TelluSYSTER and GAARAnti project (ANR-17-CE31-0009). DJJvH acknowledges Netherlands Organization for Scientific Research (NWO) Vici grant 865.17.001.

**Competing interests:** The authors have declared that no competing interests exist.

now, paleontological findings of chinchilloid rodents [2, 3] and eleutherodactylids [4] have identified that part of this colonization occurred in late Eocene to early Oligocene times (∼40–30 Ma). This is consistent with relaxed molecular clock constraints from terrestrial faunal elements from the Caribbean such as sloths (recently extinct [5, 6]) and spiders [7], amphibians such as eleutherodactylid frogs and bufonids [8, 9], and freshwater aquatic organisms such as cichlids [10, 11].

This is surprising, because these islands are separated from the South American continent by the mostly oceanic, and deep-marine Caribbean plate. This intriguing colonization inspired the hypothesis that dispersals occurred through emerged areas forming a contiguous or semi-contiguous land bridge: the hypothesis of 'GAARlandia', the land of Greater Antilles and Aves Ridge [12, 13]. Indeed, the Paleocene-Late Eocene Grenada Basin separates the Aves Ridge from the Lesser Antilles arc where Cretaceous to Paleogene volcanic and plutonic rocks of island arc affinities occur, thus Itturalde-Vinent and McPhee [12] postulate that the Aves ridge and the Lesser Antilles consisted once in a single volcanic arc connected with the Aruba-Tobago belt to the south and the Greater Antilles to the north. According to these authors, the synchronous cessation of volcanic activity along Aves and the opening of the Grenada Basin might reflect a local change in the subduction regime. Today, the Aves Ridge is a submarine high, but still submerged to ∼1km depth, formed by a former, Mesozoic to possibly Paleogene volcanic arc, and most islands of the Lesser Antilles are post-Eocene volcanoes separated by deep-marine basins [14, 15] (Fig 1A). Because the land bridge has so far not been demonstrated from the geological or geophysical record, and no mechanisms have been shown to explain its rise and demise, the alternative hypothesis would be over-water dispersal through flotsam [16].

Potential mechanisms for uplift and subsidence of a land bridge include intra-plate deformation, volcanism, and dynamic topography. In that respect, it is interesting that in the Eocene epoch leading up to the rise of the conceptual land bridge, the American and the Caribbean plates where involved in a major relative and absolute plate motion change, leading to a tectonic reorganization of their boundaries (e.g., [17–21]). In a mantle reference frame, the Caribbean Plate had until the mid-Eocene been slowly moving NE towards North America, whilst North America moved SW wards. Between 50–40 Ma, North American absolute plate motion changed to westwards [20, 21], the Caribbean Plate became near mantle stationary, and a new plate boundary formed along the E-W trending Motagua-Cayman transform system, cutting off Cuba and the Yucatan Basin from the Caribbean Plate [18, 19] (Fig 1A). Around the same time, the buoyant Bahamas bank entered the subduction zone below Cuba and Hispaniola, uplifting these two Greater Antillean islands, solving part of the GAARlandia hypothesis [12] (Fig 1). Puerto Rico followed in the late Eocene-early Oligocene [22], but the northern Lesser Antilles and the Aves Ridge were away from, and not significantly affected by this collision (Fig 1B). However, due to this plate reorganization, from the middle Eocene onwards, highly oblique subduction below the Lesser Antilles became trench-normal subduction, i.e. at much higher net subduction rates, which may have contributed to regional deformation and uplift and crustal growth through magmatism (Fig 1C). The tectonic evolution of the northeastern Caribbean region is thus of interest to evaluate the plausibility of the GAARlandia hypothesis.

Here we evaluate to what extent mid-Eocene changes in plate kinematic setting generated intra-plate deformation that may have contributed to land uplift and emergence in the northern Lesser Antilles islands. Most of these islands are Miocene and younger volcanoes, but the northeastern ones, between the active volcanic arc and the trench, expose Eocene volcano-sedimentary rocks [23, 24] that reveal evidence for Caribbean plate deformation [23, 25]. We study the Eocene deformation history based on the onshore geological record of

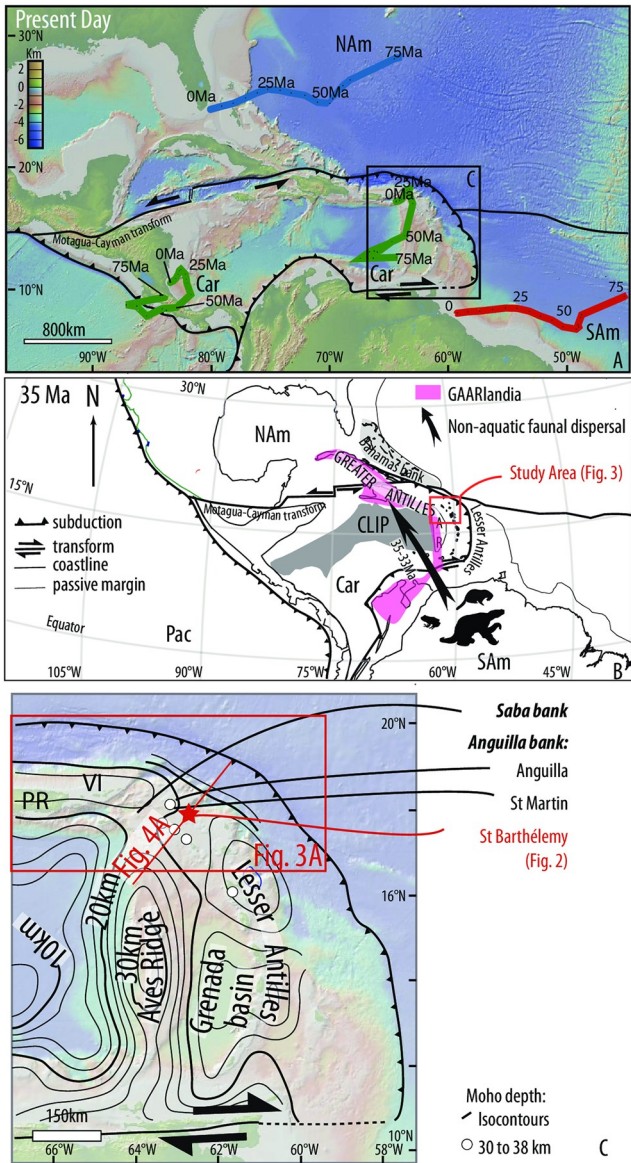

**Fig 1. Geodynamics of the Caribbean plate.** A) Present-day. The North, South American and Caribbean Plate motions are provided in blue, red and green, respectively, in the hot spot reference frame [27]. B) 35 Ma after [20]. The GAARlandia land bridge, as drawn by [12] and [2], is indicated in light pink; the arrow indicates the dispersal of South American non-aquatic animals toward the Greater Antilles. PR, VI, AR, GB and CLIP, Car, Sam, Nam stand for, Puerto-Rico, Virgin Island, Aves Ridge, Grenada Basin, Caribbean Large Igneous Province, Caribbean plate, South American plate and North America plate, respectively. C) Map of the Lesser Antilles showing the Aves Ridge, Grenada Basin, Lesser Antilles and crustal thickness (iso-contours from gravity modeling after [28, 29]; white dots: Moho depth estimates based on receiver function inversions from [30, 31]). Figure made with GeoMapApp (www.geomapapp.org) / CC BY [32].

St. Barthélemy Island combined with offshore seismic profiles orthogonal to the Lesser Antilles trench that were collected during the GARANTI cruise [26]. We evaluate our findings in terms of the occurrence, timing, and style of intra-plate deformation, and estimate whether this may have contributed to the rise and demise of a landmass that would have connected the Greater Antilles to the Aves Ridge.

## Geodynamic history and geological settings

The Caribbean Plate consists mainly of Jurassic to Lower Cretaceous oceanic crust derived from the Farallon Plate [33]. Its lithosphere is partly covered by the ∼90 Ma Caribbean Large Igneous Province [14, 34], which lies west of an extinct intra-oceanic arc represented by the Aves Ridge and an associated Jurassic back-arc basin to the east that currently underlies the Lesser Antilles islands [14, 15, 17] (Fig 1C). The plate migrated northeastward between the Americas during Cretaceous to early Eocene times, accommodated by southwest-dipping subduction below the Greater Antilles arc now found on Cuba, Hispaniola, and Puerto Rico [17], and a propagating transform plate boundary, or a highly oblique subduction zone along the eastern Caribbean plate boundary. During the 50–40 Ma plate reorganization, subduction became near-trench-normal at the Lesser Antilles trench [20]. In the upper plate, the oceanic Grenada back-arc basin started forming in the southeastern Caribbean region [28, 35], while small-scale field evidence has been presented suggesting that the northeastern Caribbean region underwent shortening [23, 36]. The Aves Ridge lies in the present-day back arc region of the Lesser Antilles active arc. From north to south, there is a conspicuous morphologic dichotomy: between the Aves Ridge and the active arc in the south lies the Grenada basin with a flat bathymetry, a mean depth of 2800 m, and a shallow ∼10–15km depth Moho. In the north, however, the seafloor topography is rough, its mean bathymetry is -1400 m, and the Moho is deeper, ~25 km [28, 29](Fig 1C). The Moho beneath the present-day arc and the fore-arc even reaches 29–38 km depth in northern Lesser Antilles [30, 31, 37]. (Fig 1C). Such a thick crust there is puzzling, and may have even been thicker given that the northern Lesser Antilles region accommodated extension since at least the Oligocene–Miocene interval [23, 38, 39]. Remnants of Eocene–Oligocene magmatic rocks and synchronous and younger sediments are exposed in the northeastern Lesser Antilles forearc, whereas most other islands expose upper Miocene to recent active arc magmatic rocks. These Eocene–Oligocene volcanic rocks were often formed subaerially, and erosional unconformities have been documented above Paleocene–Eocene marine series affecting the Anguilla and Saba banks [23, 24, 40]. This suggests that during and after the Eocene Caribbean plate reorganization, the northeastern Caribbean region underwent vertical motions and emergence, followed by subsidence, which is of interest in evaluating the GAARlandia hypothesis.

## Eocene thrusting in the Lesser Antilles

### Onshore deformation

St. Barthélemy Island (NE of the Caribbean Plate, Fig 1C) exposes middle Eocene to lower Miocene interbedded volcanoclastic rocks and limestones, both intruded by shallow plutons (Fig 2A) [23, 24]. The regional bedding is monoclinal with a mean dip of 20˚toward the S-SE, with local perturbations in the vicinity of intrusions or normal faults [23]. In the eastern part of the island, however, we mapped previously unrecognized folds and faults. There, a volcano-sedimentary and carbonate series that is tilted ∼70˚W-WSW is separated by a hanging wall cutoff from a 10˚SSW-dipping footwall (Fig 2B). The contact consists of a brittle shear zone with striations indicating westward displacement (Fig 2B and 2C). We interpret this structure as a thrust ramp-and-flat system with a top-to-the-W displacement, indicating a flat part located offshore to the east, crosscut by granodioritic intrusion as well as by younger, steeply dipping normal faults (Fig 2D). The hangingwall limestone beds consist of wackestones with large benthic foraminifera (S1 Fig) (e.g., *Coleiconus christianaensis* Robinson, S1Aa and S1B Fig, and *Barattolites trentinarensis* Vecchio and Hottinger), indicating early to mid-Eocene ages -Ypresian to Lutetian/Bartonian- [24, 41]. The occurrence of rotaliids such as *Medocia* sp.

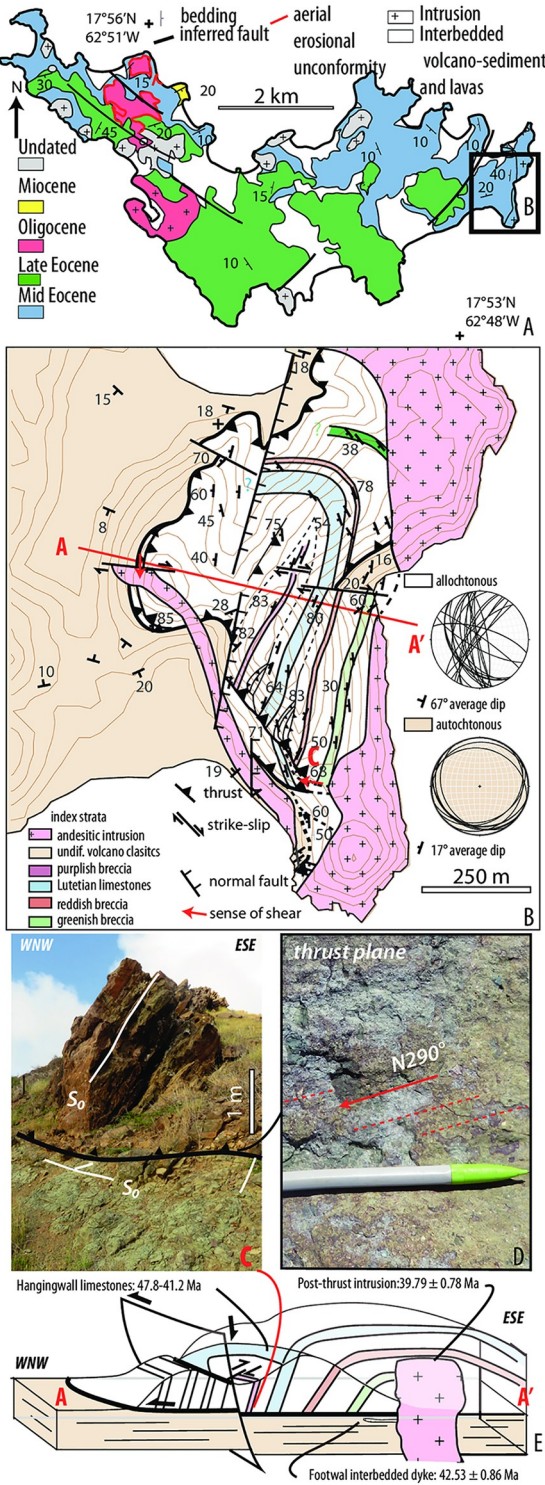

**Fig 2. St. Barthélemy.** A) Simplified geological map modified after [23, 24]; B) High-resolution map of the eastern part of the island, stereograms shows the average dip in the footwall (autochthonous) and hangingwall (allochthonous) of the thrust; radioisotopic and biostratigraphic age are indicated. Field photograph of C) the highly tilted allochthonous lying on the sub-horizontal autochthonous and D) shows the thrust with striation indicating a WSW directed thrusting, E) ~ E-W cross section AA' of the thrust.

(S1Ab Fig), miliolids such as *Praerhapydionina* (S1Cb Fig) and planktonic foraminifera, such as *Turborotalia pessagnoensis* (Tourmakine and Bolli) and *Planorotalites pseudoscitula* (Glaessner) (S1Cc and S1D Fig), further constrains the age to the Lutetian (P10-P12a, 47.8-41.2 Ma; [42, 43]). The post-thrust magmatic intrusion was dated at 39.79 ± 0.78 Ma (40Ar/39Ar on feldspar, sample SB16-10; Legendre et al., 2018). An andesitic dyke exposed in the footwall immediately below the thrust (Fig 2B and 2D, yielded an age of 42.53 ± 0.86 Ma (40Ar/39Ar on feldspar, S1 File). Thus, the time range for thrusting is constrained between *ca* 42.5 Ma and ca 39.8 Ma (with 2$\sigma$ level errors).

## Spatial extension of the thickened domain

Offshore seismic profiles and gravity data were acquired during the GARANTI cruise west of St. Barthélemy, around and on the Saba Bank, in the Lesser Antilles arc, and backarc area (S2 File) (Fig 3). We provide a 2D model of gravity data (S3 File) that gives a 40 km crustal thickness below the Lesser Antilles forearc, which is consistent with previously published data (30, 31)(Fig 4A and 4B). We correlated the seismic stratigraphy correlated with the stratigraphy constrained in the SB2 petroleum well [40] (Fig 3A). A major erosional unconformity separates a upper Eocene and younger retrograding sedimentary sequence (34.7 and 38.4 Ma andesites sit at the base of this sequence; [40]), from an inverted Paleocene to mid-Eocene basin, whose basement is inferred to be Late Cretaceous in age. Our 3D block model is constrained by land-sea correlation of seismic lines with field-mapping and shows the regional deformation pattern characterized by imbricated thrust slices and a west-verging thrust system (Fig 3B). We have reconstructed displacements along the steep dipping normal faults (Fig 4C), unfolded the thrust sheet along the ramp-flat system toward the trench and also restored lower crustal shearing (Fig 4D).We estimate a bulk shortening of 25%, which for our section amounts to. ~75 km (Fig 4D). Previous estimates suggested ≥15% of Oligocene and younger extension in the northern Lesser Antilles region [23, 38].

## Discussion

Crustal shortening and thickening, and subsequent extension and thinning are expected to lead to uplift followed by subsidence in the time interval of the hypothesized GAARlandia land bridge. Correcting for this thickening would reduce the crustal thickness from the fore-arc to the Aves Ridge to approximately 30km, on par with the thickness of the crust in the southern Lesser Antilles arc where no evidence for shortening exists and that likely underwent only magmatic thickening [37] (Fig 4C).

We infer that the geodynamic cause of our reconstructed overriding plate shortening is regional, restricted to the NE Caribbean plate. On the Greater Antilles islands to the west, Cenozoic shortening prior to and during the Eocene collision with the Bahamas platform was restricted to accretionary orogenesis, stacking sedimentary units belonging to the subducting North American Plate below the overriding Caribbean lithosphere [12, 44]. On Cuba, latest Cretaceous metamorphic core complexes did not undergo inversion (e.g., [45]), suggesting that collision was not associated with major upper plate shortening, and uplift was instead dominated by underthrusting of buoyant accretionary prism rocks and Bahamas crust. In the southeast Caribbean region, upper plate deformation was extensional instead of compressional, resulting in Paleogene opening of the Grenada basin (e.g., [46]), and there is little evidence for major inversion. We infer that the shortening of the northeastern Caribbean plate is probably a response to the change in the North American absolute plate motion and the related Caribbean plate reorganization. This reorganization changed the absolute motion of North America from southwest to west ([20], Fig 5), which also affected its subducted portions.

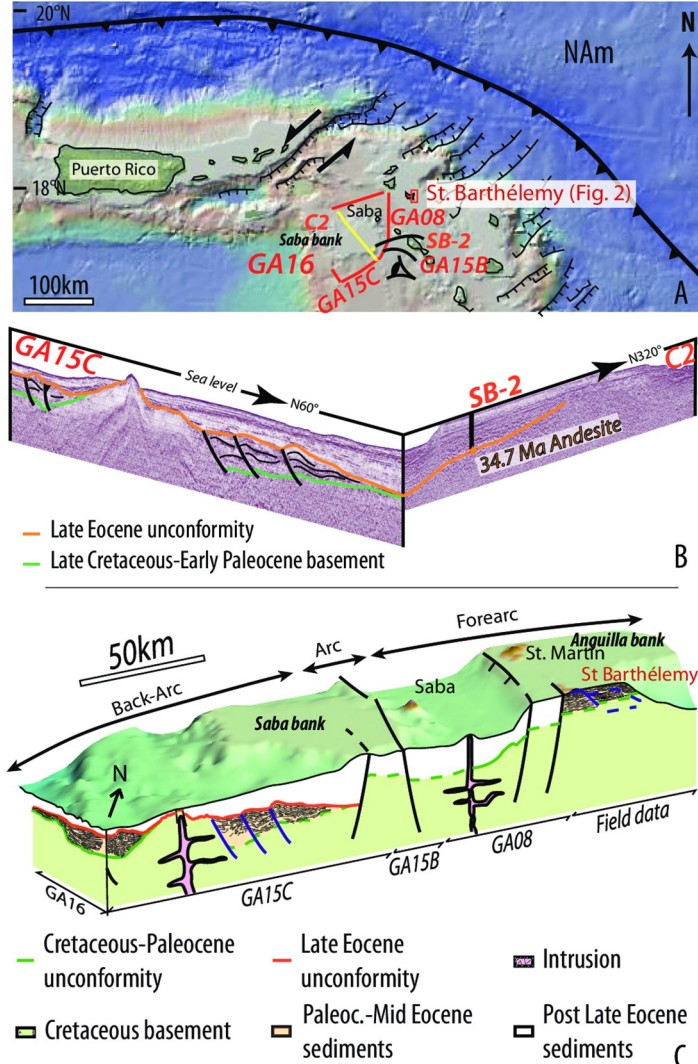

**Fig 3. Onshore-offshore geological correlation.** A) Map of the Northeastern Caribbean plate showing the location of the GARANTI Cruise seismic lines and the. Onshore-offshore geological correlation B) Correlation between the lines GA15C and C2 showing the upper Eocene regional unconformity (Orange) sealing the Eocene inverted basin. Inset: location of St. Barthélemy and seismic lines; C) Strain pattern of the Northern Lesser Antilles forearc-arc–backarc integrating land-sea data (for correlation between all the seismic data please refer to the Fig 1 provided in S2 File). The bathymetry has been extracted from the GEBCO 2014 Grid, version 20141103, GEBCO website.

At the Lesser Antilles trench, which became near-mantle stationary [20], the change resulted in trench-normal, regular subduction, but in the Puerto Rico region in the north, the change must have induced westward motion of subducted lithosphere, so-called slab dragging [47, 48]. We infer that northeastern Caribbean E-W shortening is a response of the junction region between the N-S and E-W portions of the plate boundary to this change in absolute motion of the subducting North American plate. This may for instance have led to a local flattening of the slab that was previously proposed to explain the westward migration of the active arc [49]. Late Eocene intra-plate shortening straightforwardly explains the uplift leading to the observed aerial erosional surface [23, 24] and the subsequent regional hiatus spanning the late Eocene

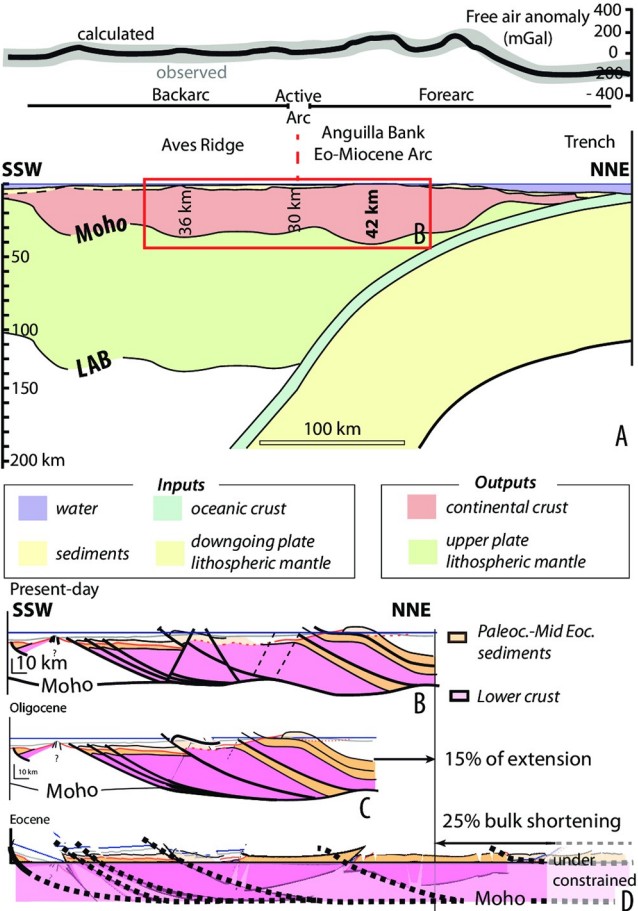

**Fig 4. Crustal strain.** A) Model of the crustal thickness (inversion of gravimetric data) along a NE-SW trending line from the trench to the Venezuela basin (location Fig 1C); B) Crustal strain pattern of the Northern Lesser Antilles; C) Restoration of the extensive deformation; D) Restoration of the compressive deformation.

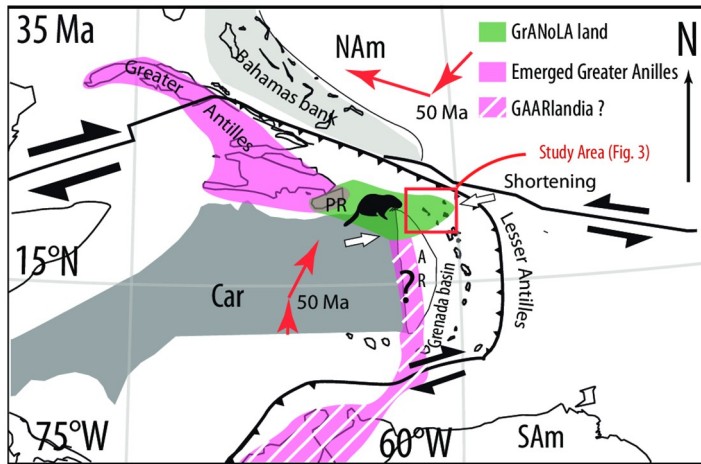

**Fig 5. Paleogeography of the northern Lesser Antilles realm at 35 Ma.** Map of the eastern Caribbean at 35 Ma modified after [20], showing the domain affected by upper crustal shortening. GrANoLA stands for 'Greater Antilles-Northern Lesser Antilles'.

and much of the Oligocene, thereby supporting the hypothesis of an intra-oceanic subaerial connection between Puerto Rico and the northern Lesser Antilles ('GrANoLA' land, i.e. Greater Antilles-Northern Lesser Antilles, Fig 5), which is fully consistent with the antiquity and phylogenetic affinities of fossil chinchilloid rodents from the concerned islands (early Oligocene–Holocene; *Borikenomys praecursor* and 'giant hutias'; [3]). Post-Oligocene demise of this landmass is reflected in a regional Miocene transgression due to regional subsidence, and is straightforwardly explained by trench parallel forearc extension and forearc block rotation in combination with thermal relaxation related to the cessation of arc activity in the northeastern Lesser Antilles islands [25, 39, 50].

Yet, the uplift and emergence of the northeastern Caribbean region add credibility to the GAARlandia hypothesis as it supports the presence of landmasses larger than the ones currently present in the region. Nevertheless, regional topography south of GrANoLA is not satisfactorily constrained yet and thus it remains unclear whether a continuous landspan or closely-spaced islands have occurred there by mid-Cenozoic times, as suggested by [13]. In any case, this topographic pattern and the short temporal duration of this event have most likely acted as a filter for Paleogene terrestrial vertebrates. Moreover, our results shed new light on the tectonic and geodynamic changes that affected the eastern Caribbean region and the area of the hypothesized land bridge during the late Eocene biological dispersion. This provides an urgency for further study of the potential causes and locations of land bridge formation and demise of a connection of to South America, in the Aves Ridge or Lesser Antilles arc and forearc region.

Our work connects geodynamic changes underlying the Caribbean plate reorganization to regional paleogeographic evolution which is reflected in current biology. The geodynamic change leading to and resulting from the plate reorganization may provide context to search for the connection of GrANoLA to South America through the Aves ridge or the Lesser Antilles arc. We identify the causes and evolution of the Grenada basin though to be Eocene in age [20, 28, 35, 46, 51, 52] (Fig 5), as the next frontiers in finding a geodynamic explanation for the rise and demise of the alleged GAARlandia faunal land bridge.

## Conclusion

The intriguing dispersals of terrestrial South American faunae to the Greater Antilles islands in the northern Caribbean region are thought to have occurred over a so far undocumented 'GAARlandia' land bridge. We perform a land-sea survey combined with chronostratigraphic data in the NE Caribbean region, which reveals a regional episode of mid-Eocene, trench-normal $\sim$E-W intra-plate crustal shortening ($\pm$75 km) and thickening (up to 40 km) that affected the northern Lesser Antilles realm. This episode coincides with a regional late Eocene–Oligocene hiatus in the Lesser Antilles that reveals uplift of a landmass between Puerto Rico and northern Lesser Antilles and lends credibility to the GAARlandia hypothesis. We tentatively explain this deformation as a response to a well-documented North American absolute plate motion change and a reorganization of the Caribbean plate boundary. Our results now direct attention to the connection from the NE Caribbean region to South America, through the Aves Ridge and/or the Lesser Antilles arc, and provide geodynamic context for intra-plate deformation and vertical motions in the time window of South American faunal dispersal to the northern Caribbean region.

## Supporting information

**S1 Fig. Photomicrograph of the stratigraphical significant foraminifera found in Pointe Toiny limestones.** Scale bars = 1mm. A) a) *Coleiconus christianaensis* (Robinson), b) *Medocia*

*sp.*, c) *Textularia sp.*, sample PON8. B) *Coleiconus christianaensis* (Robinson), sample PON1. C) a) *Halimeda sp.*, b) *Praerhapydionina sp.*, c) *Turborotalia pessagnoensis* (Tourmakine and Bolli), sample PON4. D *Planorotalites pseudoscitula* (Glaessner), sample PON1. (DOCX)

**S1 File. Geochronology: 40Ar/39Ar dating.**
(DOCX)

**S2 File. Marine geophysics: GARANTI cruise data acquisition and processing.**
(DOCX)

**S3 File. Gravity data modeling method.**
(DOCX)

## Acknowledgments

The authors thank Saba Bank Resources N.V. and managing director Clark Gomes-Casseres for the provision of Saba Bank 2 well data. Arthur Iemmolo is thanked for technical support during Ar/Ar analyses. The authors thank J. Collier, C. Montes, J. Ali, R. Mc Phee and an anonymous reviewer for their constructive comments that greatly improve our manuscript. G. Maincent, O. Maincent, H. Bernier and St Barth Essentiel are warmly thanked for their logistic support and their kindness. GARANTI Team membership list: Agranier, A., Arcay, Aude-mard, F., Beslier, M.O., Boucard, M.,Cornée, J.J., Fabre, M., Gay, A., Graindorge, D., Klingel-hoefer, A. Heuret, F., Laigle, M., Lallemand, S., Léticée, J.L., Malengro, D., Marcaillou, B., Mercier de Lepinay, B., Münch, P., Oliot, E., Oregioni, D., Padron, C., Quillévéré, F., Ratzov, G., Schenini, L., Yates, B., Lallemand, S., and Lebrun, J.F.

Lead authors of the GARANTI Team: Jean-Frederic.Lebrun@univ-antilles.fr (JFL); lallem@gm.univ-montp2.fr (SL).

## Author Contributions

**Conceptualization:** Mélody Philippon, Jean-Jacques Cornée, Philippe Münch.

**Data curation:** Mélody Philippon, Jean-Jacques Cornée.

**Formal analysis:** Mélody Philippon, Jean-Jacques Cornée, Philippe Münch, Marcelle BouDagher-Fadel, Lydie Gailler, Fredéric Quillevere.

**Funding acquisition:** Philippe Münch.

**Investigation:** Mélody Philippon, Jean-Jacques Cornée, Philippe Münch, Douwe J. J. van Hinsbergen, Marcelle BouDagher-Fadel, Lydie Gailler, Lydian M. Boschman, Frédéric Quillevere, Leny Montheil.

**Methodology:** Mélody Philippon, Jean-Jacques Cornée, Jean Frédéric Lebrun, Serge Lallemand.

**Project administration:** Philippe Münch.

**Resources:** Philippe Münch, Marcelle BouDagher-Fadel, Lydie Gailler, Fredéric Quillevere, Aurelien Gay, Jean Fredéric Lebrun, Serge Lallemand.

**Validation:** Laurent Marivaux, Pierre-Olivier Antoine.

**Visualization:** Mélody Philippon.

**Writing – original draft:** Mélody Philippon, Jean-Jacques Cornée, Philippe Münch, Douwe J. J. van Hinsbergen, Marcelle BouDagher-Fadel, Lydie Gailler, Lydian M. Boschman, Frédéric Quillevere, Leny Montheil, Aurelien Gay, Jean Frédéric Lebrun, Serge Lallemand.

**Writing – review & editing:** Mélody Philippon, Jean-Jacques Cornée, Philippe Münch, Douwe J. J. van Hinsbergen, Marcelle BouDagher-Fadel, Lydie Gailler, Lydian M. Boschman, Frederic Quillevere, Leny Montheil, Aurelien Gay, Jean Frederic Lebrun, Serge Lallemand, Laurent Marivaux, Pierre-Olivier Antoine.

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
