## [Decision Letter · Decision Letter 0]

26 Aug 2020

PONE-D-20-20355

Eocene intra-plate shortening responsible for the rise of a fauna pathway in the northeastern Caribbean realm

PLOS ONE

Dear Dr. Philippon,

Thank you for submitting your manuscript to PLOS ONE. After careful consideration, we feel that it has merit but does not fully meet PLOS ONE’s publication criteria as it currently stands. Therefore, we invite you to submit a revised version of the manuscript that addresses the points raised during the review process.

We look forward to receiving your revised manuscript.

Kind regards,

Luca Pandolfi

Academic Editor

PLOS ONE

Additional Editor Comments:

Dear Authors,

two reviewers evaluated your manuscript and both suggested minor revisions. Please check and follow the detailed reviewer' comments about your manuscript and provide a detailed reply for any rebuttal.

Looking forward to receive your revised version.

Sincerely,

Luca Pandolfi

'Financial support has been provided by INSU TelluSYSTER and GAARAnti project (ANR-17-CE31-0009). DJJvH acknowledges Netherlands Organization for Scientific Research (NWO) Vici grant 865.17.001.'

a. Please remove any funding-related text from the manuscript and let us know how you would like to update your Funding Statement. Currently, your Funding Statement reads as follows: 'NO'

3. We note that Figures 1, 2, 3 and 5 in your submission contain map/satellite images which may be copyrighted.

a. You may seek permission from the original copyright holder of Figures 1, 2, 3 and 5 to publish the content specifically under the CC BY 4.0 license. 

4. One of the noted authors is a group or consortium; the GARANTI Team.

In addition to naming the author group, and listing the individual authors and affiliations within this group in the acknowledgments section of your manuscript, please also indicate clearly a lead author for this group along with a contact email address.

Reviewers' comments:

Reviewer's Responses to Questions

**Comments to the Author**

1. Is the manuscript technically sound, and do the data support the conclusions?

Reviewer #1: Yes

Reviewer #2: Yes

2. Has the statistical analysis been performed appropriately and rigorously? 

Reviewer #1: Yes

Reviewer #2: Yes

3. Have the authors made all data underlying the findings in their manuscript fully available?

Reviewer #1: Yes

Reviewer #2: Yes

4. Is the manuscript presented in an intelligible fashion and written in standard English?

Reviewer #1: Yes

Reviewer #2: Yes

5. Review Comments to the Author

Reviewer #1: The authors provide critical new observations on the hypothesis framed by Iturralde-Vient (and MacPhee (1999) that a terrestrial corridor of some kind briefly linked South America with island groups in the present Caribbean Sea ~34 Ma. This work concentrates on one part of the story: evidence for uplift and emergence in the northern Lesser Antilles at the end of the Eocene. Their results are consistent with earlier work, and such corroboration is important. The missing part of the puzzle, as the authors recognize and will hopefully explore further in additional contributions, concerns how or even whether GRANOLA was related to the uplift and emergence of the Aves Rise in order to complete the landspan hypothesized in the original version of the GAARlandia hypothesis. The jury is still out on that point.

I want to emphasize to the editors of this journal that this work is properly multidisciplinary in its significance, as the subject matter is very much of interest to paleobiologists. The original GAARlandia proposal has been both criticized and supported by biologists using different data sets and means of inference. The Phillipon et al. paper adds important new information about the reorganization of the terrestrial realm in the end-Paleogene Caribbean Sea, which geologists and biologists alike will find intriguing. This paper also identifies important constraints on timing and correlated tectonic events.

I have only one recommendation. In fairness, Iturralde-Vinent and MacPhee (1999) actually provided a detailed tectonic model within the context of the then-understood geological framework and evolution of the Caribbean Plate. This modelm was the basis for the reconstruction of GAARlandia´s paleogeographic scenario, whereas the text of the present paper largely overlooks this and even seems to imply that it was largely derived by paleobiogeographical considerations. Of course paleogeography and paleobiogeography overlapped: this is why the West Indies are such an interesting problem to both disciplines.

Here are some relevant passages:

“The Cretaceous and Paleogene volcanic and plutonic rocks of island arc affinities occur in Aves Ridge [AR] (Bunceet al., 1970; Fox et al., 1971; Nagle, 1972; Bouysse et al., 1985; Westercamp et al., 1985; Holcombe et al., 1990), as do Mesozoic and Eocene volcanic rocks in Lesser Antilles LA (fig. 15). This basic compositional similarity suggests that, from Cretaceous through Eocene time, AR and LA [Lesser Antilles] were a single entity: the AR–LA Volcanic Arc (Pinet etal., 1985; Bouysse et al., 1985). This arc was presumably linked geologically to the Aruba/Tobago Belt in the south and the eastern Greater Antilles in the north, because all of these landmasses possess a similar Cretaceous volcanic arc-ophiolite basement.”

“If AR and LA once comprised a single arc, it can be concluded that, at some time in the past, the GB that now separates these two entities did not exist. However, the age of this basin has not been well constrained. Inconclusive seismic evidence suggests that Granada Basin GB is filled by sedimentary rocks of Paleocene (?) to Recent age (Pinet et al., 1985; Bouysse et al., 1985; Bird, 1991), while dredge hauls from the basin’s margins consist of mostly Eocene and younger sedimentary and volcaniclastic rocks (fig. 21).” [See also fig. 21 in Iturralde-Vinent and MacPhee (1999)].

“According to Pindell (1994), GB opened between the Paleocene and Late Eocene, but we postulate a somewhat younger date (Late Eocene or younger) for the following reasons. If GB is interpreted as a back-arc basin, the disjunction of the AR–LA arc into two independent geological units (Aves Ridge remnant arc and Lesser Antilles active arc) would have probably been caused by a local change in the subduction regime (e.g., alteration of angle of dip of lower slab, or migration of position of subduction zone). We hypothesize that this event was correlated with Late Eocene cessation of volcanic activity in AR (and a concomitantly great increase in activity in LA) and increased thickness of Oligocene and younger sediments in GB (see seismic sections in Nemec [1980] and Pinet et al. [1985])."

Obviously the present paper provides a much more thorough examination of the relevant evidence than we could have provided 20 years ago, but since the history of ideas is important in science, we request that the authors recognize the insights of the GAARlandia model as well as its limitations (such as the still-ambiguous evidence for linkage between northern GAARlandia and the eastern end of the Greater Antilles).

A minor point: In fig. 5, Puerto Rico is colored as part of GRANOLA, which is fine, but it was in fact included in GAARlandia as well. Can this be shown by overlapping coloration?

Some minor, mostly grammatical points are commented on in the text.

Reviewer #2: This is a very interesting manuscript and I am glad to have had the opportunity to review it. Overall the manuscript is well organized and the data is presented properly and analyzed thoroughly with no major issues. I only have a number of minor suggestions (listed below), mainly from a grammatical point of view and some regarding the content, which I hope the authors take into consideration. The figures are very well done and informative.

General comments:

1) One point that might be added in the discussion is that although GrANoLa does not confirm the presence of a full landspan, it does confirm and support the presence of landmasses larger than the ones currently present in the region. This in turn could support a slightly distinct view mentioned by Iturralde-Vinent (2006), that instead of a continuous landspan, it consisted of a number of closely spaced islands separated by shallow marine channels. The presence of closely-spaced islands in place of a continuous landspan, plus the short temporal duration of this event, could have acted as a filter, thus accounting for the relatively low taxonomic richness of terrestrial vertebrates in Paleogene deposits in the region.

Iturralde-Vinent, M. A. 2006. Meso-Cenozoic Caribbean Paleogeography: implications for the historical biogeography of the region. International Geology Review 48:791–827.

MacPhee, R. D. E., and M. A. Iturralde-Vinent. 2000. A short history of Greater Antillean land mammals: biogeography, paleogeography, radiations, and extinctions. Tropics 10:145–154.

2) A good part of the references with geographic names need to be capitalized properly.

Specific comments:

Line 22: I suggest changing: “... respect, is interesting...:” to “... respect, it is interesting...”

Line 34: I suggest changing: “... in the Late Eocene-Early Oligocene…” to “... in the late Eocene-early Oligocene…” Late and Early are no longer used as formal subdivisions of the Eocene and Oligocene and should not be capitalized.

Line 57: typo, delete “lies west of.”

Line 95: I suggest changing: “... a hangingwall…” to “... a hanging wall…”

Line 105: I suggest changing: “... Turborotalia possagnoensis…” to “... Turborotalia pessagnoensis…”

Figure 1 caption (page 3): I suggest changing “C) shows a Map of the …” to “C) Map of the …”

Figure 4 caption (page 5): “D)” is missing.

6. PLOS authors have the option to publish the peer review history of their article (what does this mean?). If published, this will include your full peer review and any attached files.

Reviewer #1: **Yes: **Ross D. E. MacPhee

Reviewer #2: No

---

## [Author Response · Author response to Decision Letter 0]

29 Sep 2020

Reviewer #1: Ross MacPhee

The authors provide critical new observations on the hypothesis framed by Iturralde-Vient (and MacPhee (1999) that a terrestrial corridor of some kind briefly linked South America with island groups in the present Caribbean Sea ~34 Ma. This work concentrates on one part of the story: evidence for uplift and emergence in the northern Lesser Antilles at the end of the Eocene. Their results are consistent with earlier work, and such corroboration is important. The missing part of the puzzle, as the authors recognize and will hopefully explore further in additional contributions, concerns how or even whether GRANOLA was related to the uplift and emergence of the Aves Rise in order to complete the landspan hypothesized in the original version of the GAARlandia hypothesis. The jury is still out on that point.

I want to emphasize to the editors of this journal that this work is properly multidisciplinary in its significance, as the subject matter is very much of interest to paleobiologists. The original GAARlandia proposal has been both criticized and supported by biologists using different data sets and means of inference. The Phillipon et al. paper adds important new information about the reorganization of the terrestrial realm in the end-Paleogene Caribbean Sea, which geologists and biologists alike will find intriguing. This paper also identifies important constraints on timing and correlated tectonic events.

We thank the reviewer for his very positive comments.

I have only one recommendation. 

In fairness, Iturralde-Vinent and MacPhee (1999) actually provided a detailed tectonic model within the context of the then-understood geological framework and evolution of the Caribbean Plate. This modelm was the basis for the reconstruction of GAARlandia´s paleogeographic scenario, whereas the text of the present paper largely overlooks this and even seems to imply that it was largely derived by paleobiogeographical considerations. Of course paleogeography and paleobiogeography overlapped: this is why the West Indies are such an interesting problem to both disciplines.

Here are some relevant passages:

“The Cretaceous and Paleogene volcanic and plutonic rocks of island arc affinities occur in Aves Ridge [AR] (Bunceet al., 1970; Fox et al., 1971; Nagle, 1972; Bouysse et al., 1985; Westercamp et al., 1985; Holcombe et al., 1990), as do Mesozoic and Eocene volcanic rocks in Lesser Antilles LA (fig. 15). This basic compositional similarity suggests that, from Cretaceous through Eocene time, AR and LA [Lesser Antilles] were a single entity: the AR–LA Volcanic Arc (Pinet etal., 1985; Bouysse et al., 1985). This arc was presumably linked geologically to the Aruba/Tobago Belt in the south and the eastern Greater Antilles in the north, because all of these landmasses possess a similar Cretaceous volcanic arc-ophiolite basement.”

“If AR and LA once comprised a single arc, it can be concluded that, at some time in the past, the GB that now separates these two entities did not exist. However, the age of this basin has not been well constrained. Inconclusive seismic evidence suggests that Granada Basin GB is filled by sedimentary rocks of Paleocene (?) to Recent age (Pinet et al., 1985; Bouysse et al., 1985; Bird, 1991), while dredge hauls from the basin’s margins consist of mostly Eocene and younger sedimentary and volcaniclastic rocks (fig. 21).” [See also fig. 21 in Iturralde-Vinent and MacPhee (1999)].

“According to Pindell (1994), GB opened between the Paleocene and Late Eocene, but we postulate a somewhat younger date (Late Eocene or younger) for the following reasons. If GB is interpreted as a back-arc basin, the disjunction of the AR–LA arc into two independent geological units (Aves Ridge remnant arc and Lesser Antilles active arc) would have probably been caused by a local change in the subduction regime (e.g., alteration of angle of dip of lower slab, or migration of position of subduction zone). We hypothesize that this event was correlated with Late Eocene cessation of volcanic activity in AR (and a concomitantly great increase in activity in LA) and increased thickness of Oligocene and younger sediments in GB (see seismic sections in Nemec [1980] and Pinet et al. [1985])."

Obviously the present paper provides a much more thorough examination of the relevant evidence than we could have provided 20 years ago, but since the history of ideas is important in science, we request that the authors recognize the insights of the GAARlandia model as well as its limitations (such as the still-ambiguous evidence for linkage between northern GAARlandia and the eastern end of the Greater Antilles).

Following the reviewer, the introduction has been reworked and now provides a better knowledge of the GAARLandia hypothesis published by Iturralde Vinente and Mc Phee in 1999.

Indeed, the Paleocene-Late Eocene Grenada Basin separates the Aves Ridge from the Lesser Antilles arc where Cretaceous to Paleogene volcanic and plutonic rocks of island arc affinities occur, thus Itturalde-Vinent and McPhee (1999) postulate that Aves ridge and the Lesser Antilles consisted once in a single volcanic arc connected with Aruba-Tobago belt and the Greater Antilles to the south and north, respectively. According to these authors, the synchronous cessation of volcanic activity along Aves and the opening of the Grenada Basin might reflect a local change in the subduction regime.

A minor point: In fig. 5, Puerto Rico is colored as part of GRANOLA, which is fine, but it was in fact included in GAARlandia as well.

Can this be shown by overlapping coloration?

Yes indeed, this has been corrected

Some minor, mostly grammatical points are commented on in the text.

All the suggestions made by the reviewer have been taken into account.

 

Reviewer #2: Anonymous

This is a very interesting manuscript and I am glad to have had the opportunity to review it. Overall the manuscript is well organized and the data is presented properly and analyzed thoroughly with no major issues. I only have a number of minor suggestions (listed below), mainly from a grammatical point of view and some regarding the content, which I hope the authors take into consideration. The figures are very well done and informative.

We thank the reviewer for his very positive comments.

General comments:

1) One point that might be added in the discussion is that although GrANoLa does not confirm the presence of a full landspan, it does confirm and support the presence of landmasses larger than the ones currently present in the region. This in turn could support a slightly distinct view mentioned by Iturralde-Vinent (2006), that instead of a continuous landspan, it consisted of a number of closely spaced islands separated by shallow marine channels. The presence of closely-spaced islands in place of a continuous landspan, plus the short temporal duration of this event, could have acted as a filter, thus accounting for the relatively low taxonomic richness of terrestrial vertebrates in Paleogene deposits in the region.

Iturralde-Vinent, M. A. 2006. Meso-Cenozoic Caribbean Paleogeography: implications for the historical biogeography of the region. International Geology Review 48:791–827.

MacPhee, R. D. E., and M. A. Iturralde-Vinent. 2000. A short history of Greater Antillean land mammals: biogeography, paleogeography, radiations, and extinctions. Tropics 10:145–154.

Following the reviewer, the discussion has been implemented with a sentence:

"The uplift and emergence of the northeastern Caribbean region add credibility to the GAARlandia hypothesis as it supports the presence of landmasses larger than the ones currently present in the region. Nevertheless, regional topography south of GrANoLA is not satisfactorily constrained yet and thus it remains unclear whether a continuous landspan or closely-spaced islands have occurred there by mid-Cenozoic times, as suggested by Itturalde-Vinente (2006). In any case, this topographic pattern and the short temporal duration of this event have most likely acted as a filter for Paleogene terrestrial vertebrates."

2) A good part of the references with geographic names need to be capitalized properly.

Geographic names have been capitalized properly.

Specific comments:

Line 22: I suggest changing: “... respect, is interesting...:” to “... respect, it is interesting...”

Corrected according to the reviewer suggestions

Line 34: I suggest changing: “... in the Late Eocene-Early Oligocene…” to “... in the late Eocene-early Oligocene…” Late and Early are no longer used as formal subdivisions of the Eocene and Oligocene and should not be capitalized.

Corrected according to the reviewer suggestions

Line 57: typo, delete “lies west of.”

Corrected according to the reviewer suggestions

Line 95: I suggest changing: “... a hangingwall…” to “... a hanging wall…”

Corrected according to the reviewer suggestions

Line 105: I suggest changing: “... Turborotalia possagnoensis…” to “... Turborotalia pessagnoensis…”

Corrected according to the reviewer suggestions

Figure 1 caption (page 3): I suggest changing “C) shows a Map of the …” to “C) Map of the …”

Corrected according to the reviewer suggestions

Figure 4 caption (page 5): “D)” is missing.

Corrected according to the reviewer suggestions

---

## [Editor Report · Decision Letter 1]

7 Oct 2020

Eocene intra-plate shortening responsible for the rise of a faunal pathway in the northeastern Caribbean realm

PONE-D-20-20355R1

Dear Dr. Mélody Philippon,

We’re pleased to inform you that your manuscript has been judged scientifically suitable for publication and will be formally accepted for publication once it meets all outstanding technical requirements.

Kind regards,

Luca Pandolfi

Academic Editor

PLOS ONE

Additional Editor Comments (optional):

Dear Authors,

I'm glad to accept the revised version of your manuscript. You taken into account all the comments and suggestions of both reviewers and modified the manuscript accordingly.

Thank you for your very interesting submission.

Sincerely,

Luca Pandolfi

PS: concerning changing in funding information, please contact the Team at plosone@plos.org
---

## [Editor Report · Acceptance letter]

9 Oct 2020

PONE-D-20-20355R1 

Eocene intra-plate shortening responsible for the rise of a faunal pathway in the northeastern Caribbean realm 

Dear Dr. Philippon:

I'm pleased to inform you that your manuscript has been deemed suitable for publication in PLOS ONE. Congratulations! Your manuscript is now with our production department. 

Kind regards, 

on behalf of

Dr. Luca Pandolfi 

Academic Editor

PLOS ONE